# ATSTrack: Enhancing Visual-Language Tracking by Aligning Temporal and Spatial Scales

## Abstract

A main challenge of Visual-Language Tracking (VLT) is the misalignment between visual inputs and language descriptions caused by the movement of targets. Previous trackers have explored many effective feature modification methods to preserve more aligned features. However, an important yet unexplored factor ultimately hinders their capabilities, which is the inherent differences in the temporal and spatial scale of information between visual and language inputs. To address this issue, we propose a novel visual-language tracker that enhances the effect of feature modification by **A**ligning **T**emporal and **S**patial scales of different input components, named as **ATSTrack**. Specifically, we decompose each language description into phrases with different attributes based on their temporal and spatial correspondence with visual inputs, and modify their features in an attribute-specific manners. Moreover, we introduce a Visual-Language token that comprises modified linguistic information from the previous frame to guide the model to extract visual features that are more relevant to language description, thereby reducing the impact caused by the differences in spatial scale. Experimental results show that our proposed ATSTrack achieves performance comparable to existing methods. Our code is provided in Supplementary Material and will be released.

## 1 Introduction

Visual-Language tracking aims to track targets based on initial bounding boxes and additional natural language descriptions. This approach could overcome the limitations of relying solely on visual modalities and thus improve the tracking performance by leveraging high-level semantic information in language descriptions Hu et al. (2023); Li et al. (2024a;b).

A major challenge of visual-language tracking is the misalignment between visual inputs and language descriptions Shao et al. (2024); Zhou et al. (2023). In most cases, language descriptions either specify the target's initial state or offer a brief summary over time. However, as the target moves, it may undergo deformation or changes in action, leading to inconsistency with the original language description. As illustrated in Figure 1(a), the target's actions and positions are continuously changing. More examples are shown in A.2. Regarding this issue, it is crucial to modify language features in order to filter out the information that does not align with the current state of the target.

Although some effective feature modification methods have been explored by previous visual-language trackers Li et al. (2023); Ma & Wu (2023); Ma et al. (2024); Zhou et al. (2023); Sun et al. (2024a), these methods tend to overlook the inherent differences in the temporal and spatial scales of information contained in different components of visual and language inputs Wu et al. (2024); Chen et al. (2023a), and fail to achieve the optimal modification effect. Specifically, visual inputs provide detailed spatial information of the entire scene, while language descriptions typically correspond to only a small portion of the image and have limited spatial scale. Conversely, language descriptions convey broader temporal context by summarizing the behavior of the target over time (e.g., moving fast or slowly), while the search feature that was used by existing trackers to modify the language feature lacks this temporal depth. As illustrated in Figure 1(b), previous trackers use visual and language features as two distinct entireties during modification, which inevitably suffer

Figure 1: Comparison with previous trackers. (a) The misalignment between language and visual inputs. Compared with (b) previous trackers, (c) our tracker has been improved in both feature extraction and modification: we utilize a token containing linguistic information to guide the extraction of visual features, and propose a fine-grained modulation module to modify language features.

from the misalignment of temporal and spatial scales. For example, when using visual features to modify the description about the target's appearance, excessive background information would inadvertently introduce interference. To address this issue, we propose a novel visual-language tracking framework that enhances the effect of language feature modification by **A**ligning the **T**emporal and **S**patial scales of different input components, termed **ATSTrack**. Specifically, we decompose language descriptions into attribute-specific phrases based on their correspondence with different visual cues. Features of different attribute then processed by the **Attribute-Specific Modification module (ASM)**, where they interacts only with the most relevant visual features, and through distinct interaction manners that were designed based on the characteristics of different attributes. This design enables more precise and interpretable feature refinement, avoiding the interference introduced by temporal and spatial difference.

Another problem caused by the spatial scale difference arises during the extraction of visual features. As mentioned above, the spatial scale of visual inputs is usually larger than languages descriptions. In previous trackers, visual features are extracted independently without the involvement of linguistic information, which can cause the visual backbone to pay unnecessary attention to those irrelevant visual details (e.g., irrelevant objects, background), while neglecting features that are related to the language descriptions. Even if the model pays sufficient attention to the target through the interaction with the template, the focus of the features it extracts (e.g., texture, edges) may still diverge from the language descriptions (e.g., color, action). To address this issue, we introduce a **Visual-Language token (VL token)** that incorporates modified linguistic information and propagates it to the visual backbone of the following frame. In such a way, the model can extract visual features that are more relevant to language descriptions with the guidance of linguistic information.

Our main contributions are summarized as follows:

- We propose **ATSTrack**, a novel visual-language tracking framework, which could enhance the effect of feature modification by **A**ligning the **T**emporal and **S**patial scales of different input components.

- We address the interference caused by the temporal and spatial misalignment between visual and language features with a **Attribute-Specific Modification module**, and enhance the cross-modality correlation by using a **Visual-Language token** that incorporates linguistic information to guide the extraction of visual features.

- The proposed ATSTrack outperforms state-of-the-art vision-language trackers. We conducted extensive experiments including ablation studies to demonstrate the effectiveness of the proposed framework and modules.

## 2 RELATED WORK

**Visual Single Object Trackers.** Single object tracking aims to locate the target in a video sequence according to the given bounding box in the first frame. Existing mainstream trackers Cai et al. (2023b); Xie et al. (2022); Guo et al. (2020); Yan et al. (2021); Cai et al. (2023a); He et al. (2023); Kim et al. (2022) typically rely on the matching between the template and the search region. MixFormer Cui et al. (2022) uses iterative mixed attention to integrate feature extraction and target information. OSTrack Ye et al. (2022) proposes a single-stream framework that can jointly perform feature extraction and relation modeling, along with an early candidate elimination module to eliminate unnecessary search region tokens.

However, these methods may face significant challenge when the appearance of the target undergoes drastic changes (i.e., rapid motion or occlusion) Huang et al. (2024), since they use only the visual information for feature relationship modeling. Some methods have focused on utilizing motion information. SeqTrack Chen et al. (2023b) models object tracking as a sequence generation task, offers a simple framework by removing the redundant prediction head and loss function. AR-Track Wei et al. (2023) treats tracking as a coordinate sequence interpretation task and uses a time autoregressive method to model changes in trajectory sequences, thereby maintaining cross-frame tracking of the target. Despite using additional motion information, these methods still heavily rely on visual matching and cannot completely eliminate the aforementioned limitation.

**Visual-Language Trackers.** Visual-Language tracking aims to track targets based on visual features and additional natural language descriptions. TNL Li et al. (2017) first introduces natural language into tracking, achieving more robust results than visual trackers. SNLT Feng et al. (2021) uses language and visual information to predict the state of the target individually and then fuses these predictions to obtain the final tracking result. VLT Guo et al. (2022) proposes modality mixer for unified Visual-Language representation learning and the asymmetric searching strategy to mix Visual-Language representation.

Recently, more researchers are beginning to notice the mismatch between visual and language inputs. DecoupleTNL Ma & Wu (2023) decouples the tracking task into short-term context matching and long-term context perceiving. QueryNLT Shao et al. (2024) proposes a multi-modal prompt modulation module to filter out information by leveraging the complementarity between visual inputs and language descriptions. Unlike other methods that rely on manual language annotations, CiteTracker Li et al. (2023) uses CLIP Radford et al. (2021) to generate initial attributes for the target and adjust the weights of these attributes in each frame. However, these methods still suffer from the temporal and spatial differences between visual and language inputs. To this end, we propose a novel framework that enhances the effect of feature modification by aligning the temporal and spatial scale of different input components.

## 3 METHOD

### 3.1 OVERVIEW

Figure 2 shows the general framework of the ATSTrack. The input of the visual backbone includes the search image, the template sequence, and the Visual-Language token from the previous frame. The output of the visual backbone consists of: search feature $F_{\text{search}}$, template features $F_{\text{temp}} = \{F_0, ..., F_{\text{n-1}}, F_{\text{n}}\}$, the $cls$ token of the visual backbone is defined as visual token $T_{\text{vi}}$. We utilize a Large Language Model (LLM) to segment each language description into four phrases with different attributes based on their correspondence with visual inputs: **Category**, **Appearance**, **Action**, and **Location**. The language backbone subsequently extracts features of these various attributes: category feature $F_{\text{cate}}$, appearance feature $F_{\text{app}}$, action feature $F_{\text{act}}$ and location feature $F_{\text{loc}}$.

These visual and language features are then fed into the Attribute-Specific Modulation module to acquire modified language features $F_{\text{lang}} = \{F_{\text{cate}}, \overline{F}_{\text{app}}, \overline{F}_{\text{act}}, \overline{F}_{\text{loc}}\}$. We generate a language token $T_{\text{lang}}$ from $F_{\text{lang}}$ and aggregate $T_{\text{lang}}$ with $T_{\text{vi}}$ as the Visual-Language token $T_{\text{VL}}$, which is propagated to the visual backbone of the next frame to guide the extraction of visual features. $F_{\text{lang}}$ and $F_{\text{search}}$ are merged through cross attention and sent to the prediction head to obtain the tracking result. Supplementary details of our design rationale are elaborated in A.3

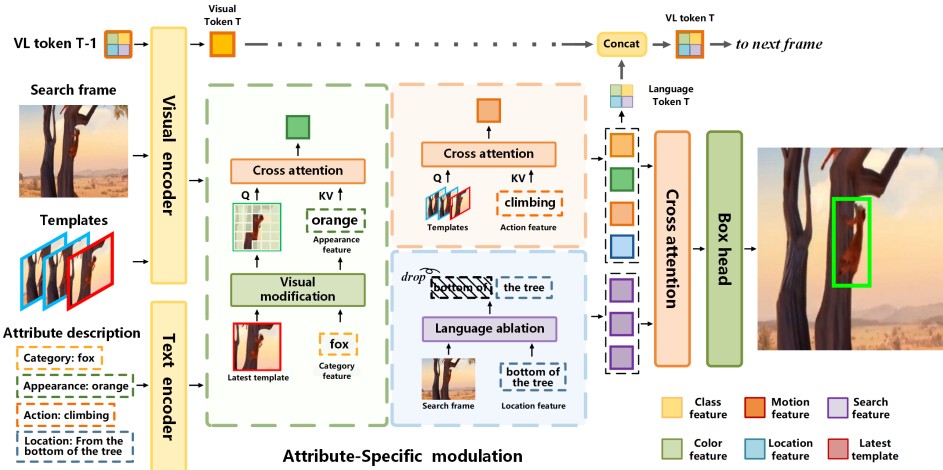

Figure 2: Overview of the ATSTrack framework. ATSTrack has been improved in two aspects: 1) A Visual-Language token is used to guide the extraction of visual features. 2) An Attribute-Specific Modulation module is designed to make more effective modification to the language features.

## 3.2 VISUAL LANGUAGE CORRESPONDENCE

As previously mentioned, we segment each complete language description into four phrases with different attributes based on their correspondence with different visual inputs: **Category**, **Appearance**, **Action**, and **Location**. For instance, "Yellow airplane flying in the air" will be segmented as {"Category: airplane", "Appearance: yellow", "Action: flying", "Location: in the air"}, more examples are shown in Figure 4. In this section, we provide a detailed explanation of the characteristics of different attributes and their correspondences with visual inputs. Details about the usage of LLM are shown in A.4.

**Category and Appearance.** "Category" and "Appearance" correspond to the latest template rather than search frame, as template contains less background and can better reflect the object's category and appearance. The category descriptions are usually accurate and require no further modification, while the appearance may vary, so we categorize them separately.

**Action.** "Action" refers to the motion state of the target. We consider that "Action" corresponds to the entire template sequence because it could be difficult to distinguish between actions such as "walking" and "running" using a single template. It should be noted that the interaction between the target and other objects is considered as "Location", as other objects may be distant from the target and thus not appear in the template.

**Location.** Descriptions of an object's location often involve other objects in the background, so "Location" should correspond to the search image. As mentioned above, "Location" includes not only the literal description of where an object is located, but also other descriptions that help locate the target, such as "played by a man".

## 3.3 ATTRIBUTE-SPECIFIC MODIFICATION

The structure of the Attribute-Specific Modulation module is shown in Figure 2. Compared to the coarse-grained interaction used by previous trackers, attribute-specific interaction enables more precise feature modification by explicitly aligning the temporal and spatial scales of different input components. Moreover, we design distinct modification strategies based on the unique characteristics of different attributes: 1) Since the category of target usually remains constant, we keep the category feature $F_{cate}$ unchanged. 2) The appearance feature is modified with the latest template in the template sequence, denoted as $F_n$. We employ a **Visual Feature Modification (VFM)** module that leverages $F_{cate}$ to suppress background information in $F_n$ to prevent interference. 3) The action feature $F_{act}$ is modified using all template features $F_{temp}$ through cross attention since they both contain rich temporal context. We set the number of templates to 3 following the setting in Zheng et al. (2024) and Li et al. (2025), detailed experiments about the effect of multiple templates are provided

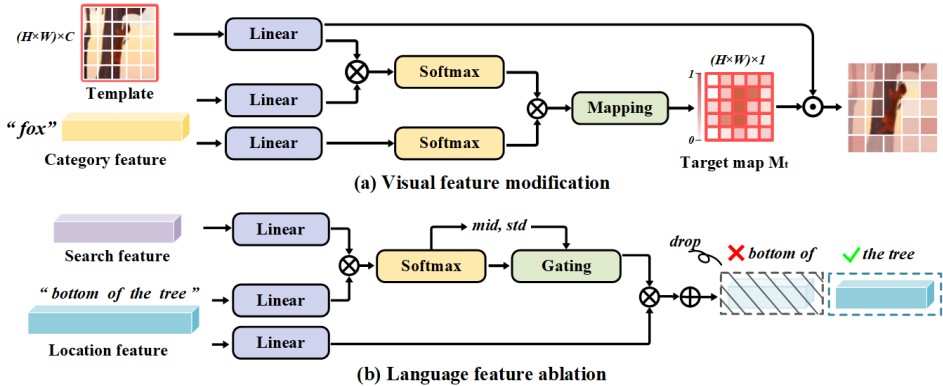

Figure 3: (a) The structure of the Visual Feature Modification module and (b) Language Feature Ablation module.

in A.5. 4) We utilize a **Language Feature Ablation (LFA)** module that could selectively removes the misaligned components in location features.

**Visual Feature Modification.** The structure of the VFM is illustrated in Figure 3 (a). Previous methods Shao et al. (2024); Li et al. (2025) typically aim to directly remove background from the template. However, since we use the template features to modify language features, we hope to suppress background while preserving small, language-related attachment of target (e.g., handbag, umbrella). Therefore, we design a more flexible approach to adjust the weight of each token in $F_{\text{n}}$. Given the category feature $F_{\text{cate}} \in \mathbb{R}^{L \times C}$ and the template feature $F_{\text{n}} \in \mathbb{R}^{(H_t \times W_t) \times C}$ as input, we adopt linear projection layers to project them to the same dimension and calculate the similarity matrix $M_{sim} \in [0,1]^{(H_t \times W_t) \times L}$ between category and template features:

$$M_{sim} = \text{softmax}\left(\frac{\delta_t(F_{\text{n}}) \times \delta_c(F_{\text{cate}})}{\sqrt{C}}\right) \tag{1}$$

where $\delta_c$ and $\delta_t$ are projection layers for category features and template features. Since the importance of the information contained in different tokens of $F_{\text{cate}}$ also varies Shao et al. (2024), we calculate the importance score map of $F_{\text{cate}}$ with softmax function and multiply it by $M_{sim}$ to increase the difference between target and background in the target map $M_t \in [\alpha, 1]^{(H_t \times W_t) \times 1}$. Finally, the modified template feature $\overline{F}_{\text{n}} \in \mathbb{R}^{(H_t \times W_t) \times C}$ is acquired by:

$$M_t = \Phi(M_{sim} \times \text{softmax}(\delta_t(F_{\text{n}}))) \tag{2}$$

$$\overline{F}_{\text{n}} = F_{\text{n}} \odot M_t \tag{3}$$

$\Phi$ is a mapping function, and we provide more detailed explanations in A.6. The values in $M_t$ reflect the probability that the features belong to the target. Through this method, we can suppress background features while retaining attention on attachments that are semantically associated with the target.

**Language Feature Ablation.** The structure of the LFA is illustrated in Figure 3 (b). Existing positional descriptions can be categorized into two forms: either general ("at location A") or detailed ("from A to B"). General form are usually very broad (e.g., on the ground) and therefore more accurate, while detailed form, due to involving too many specifics, are typically not entirely accurate at most given moment. Based on this observation, LFA should satisfy two requirements: 1) when misalignment exists, it should completely remove the misaligned components. 2) when no misalignment exists, it should preserve as much valid information as possible. Motivated by this, we design a gating mechanism based on dynamic threshold. The gating operation adjusts the values in the similarity matrix $M_{sim} \in [0,1]^{(H_s \times W_s) \times L}$ between search feature $F_{\text{search}} \in \mathbb{R}^{(H_s \times W_s) \times C}$ and location feature $F_{\text{loc}} \in \mathbb{R}^{L \times C}$, which is used as the weight to aggregate information in $F_{\text{loc}}$:

$$\theta = \text{med}\left(M_{sim}^j\right) + \varphi \text{std}\left(M_{sim}^j\right) \tag{4}$$

$$G_j = \text{sigmoid}\left(\alpha\left(M_{sim}^j - \theta\right)\right) \tag{5}$$

$$M = M_{sim} \odot G \tag{6}$$

where $\alpha = 50$, $\varphi = 0.5$. $M_{sim}^j$ is the $j_{th}$ column of $M_{sim}$. We use the weighted sum of the median and variance of $M_{sim}^j$ to initialize a threshold $\theta$, when the values in $M_{sim}^j$ are less discrete (i.e., the information among different tokens $F_{\text{loc}}$ is more consistent), $\theta$ is smaller and allows more valid information to be retained. We subtract $\theta$ from $M_{sim}^j$ and multiply it with scaling factor $\alpha$ before applying the sigmoid function to obtain $G_j$, which represents the $j_{th}$ column of gating matrix $G \in (0,1)^{(H_s \times W_s) \times L}$. The values in $G$ are directly proportional to the similarity scores in $M_{sim}$. By multiplying $G$ with $M_{sim}$, the weights of tokens in $F_{\text{loc}}$ that exhibit low similarity between $F_{\text{search}}$ will be projected to close to 0. The modified location feature $\overline{F}_{\text{loc}} \in \mathbb{R}^{L \times C}$ is acquired by:

$$\overline{F}_{\text{loc}} = M \times \delta_v(F_{\text{loc}}) + F_{\text{loc}} \tag{7}$$

where $\delta_v$ represents the projection layer for $F_{\text{loc}}$.

## 3.4 VISUAL-LANGUAGE TOKEN

The need for early cross-modal interaction has already been noticed in the field of object detection Liu et al. (2023). However, previous visual-language trackers still confine the backbone's access to information to a single modality. This overlook of cross-modality information interaction exacerbates the misalignment between visual and language features, thereby affecting the effectiveness of subsequent operations.

To address this issue, we generate a Visual-Language token $T_{\text{VL}} \in \mathbb{R}^{2 \times C}$ for each video frame and propagate it to the visual backbone of the following frame. $T_{\text{VL}}$ is the aggregation of the visual token $T_{\text{vi}} \in \mathbb{R}^{1 \times C}$ and language token $T_{\text{lang}} \in \mathbb{R}^{1 \times C}$. $T_{\text{vi}}$ is the $cls$ token of the visual backbone, which consists of the global visual information. After acquiring the modified language features $F_{\text{lang}}$, we take the global average of $F_{\text{lang}}$ as language token $T_{\text{lang}}$ and concatenate $T_{\text{vi}}$ with $T_{\text{lang}}$ to acquire the Visual-Language token $T_{\text{VL}}$. The overall process can be formulated as:

$$T_{\text{lang}} = \text{avg}\left(\text{concat}\left[F_{\text{cate}}, \overline{F}_{\text{app}}, \overline{F}_{\text{act}}, \overline{F}_{\text{loc}}\right]\right) \tag{8}$$

$$T_{\text{VL}} = \text{concat}\left[T_{\text{lang}}, T_{\text{vi}}\right] \tag{9}$$

where concat[·,·] denotes the concatenation operation. $T_{\text{VL}}$ is concatenated with visual input of the next frame. From the perspective of context understanding, $T_{\text{VL}}$ contains global visual and linguistic information from the previous frame, which helps the model to better model the temporal relationships between frames. From the perspective of visual-language alignment, the linguistic information contained in $T_{\text{VL}}$ guides the model to extract features that are more relevant to language descriptions by participating in subsequent attention operations within the visual backbone.

## 3.5 PREDICTION HEAD AND LOSS FUNCTION

We employ a commonly used prediction head Ye et al. (2022); Gao et al. (2023); Zheng et al. (2025) comprising 3 conventional branches to obtain the center score map $C \in [0,1)^{\frac{H_x}{p} \times \frac{H_x}{p}}$, an offset map $O \in [0,1)^{2 \times \frac{H_x}{p} \times \frac{H_x}{p}}$ and a normalized size map $S \in [0,1)^{2 \times \frac{H_x}{p} \times \frac{H_x}{p}}$, where $p$ is the size of the image patches. The final tracking results are computed as follows:

$$(x, y, w, h) = \text{map}\left(x_c + O_x, y_c + O_y, S_x, S_y\right) \tag{10}$$

where $(x_c, y_c) = \text{argmax}(C)$ and map$(\cdot)$ represents the operation of mapping the bounding box back to its original size.

We adopt the focal loss as classification loss $L_{cls}$, and the $L1$ loss and $GIoU$ loss as regression loss. The overall loss function can be formulated as:

$$L = L_{cls} + \lambda_1 L_1 + \lambda_2 L_{GIoU} \tag{11}$$

we set $\lambda_1 = 5$ and $\lambda_2 = 2$ following the common setting in SOT.

| Method | | Source | TNL2K | | | LaSOT | | | OTB$_{lang}$ | | |
|---|---|---|---|---|---|---|---|---|---|---|---|
| | | | AUC | P$_{norm}$ | P | AUC | P$_{norm}$ | P | AUC | P$_{norm}$ | P |
| Visual-only trackers | SwinTrack-B$_{384}$Lin et al. (2022) | NIPS22 | 55.9 | - | 57.1 | 71.3 | - | 76.5 | - | - | - |
| | OSTrack$_{384}$Ye et al. (2022) | ECCV22 | 54.3 | - | - | 71.1 | 81.1 | 77.6 | - | - | - |
| | MixFormer-v2 Cui et al. (2023) | CVPR22 | 57.4 | - | 58.4 | 70.6 | 80.8 | 76.2 | - | - | - |
| | ARTrack-B$_{384}$Wei et al. (2023) | CVPR23 | 58.9 | - | - | 72.6 | 81.7 | 79.1 | - | - | - |
| | SeqTrack-B$_{384}$Chen et al. (2023b) | CVPR23 | 56.4 | - | - | 71.5 | 81.1 | 77.8 | - | - | - |
| | DropTrack$_{384}$Wu et al. (2023) | CVPR23 | 56.9 | - | 57.9 | 71.8 | 81.8 | 78.1 | - | - | - |
| | AQATracker$_{384}$Xie et al. (2024) | CVPR24 | 59.3 | - | 62.3 | 72.7 | 82.9 | 80.2 | - | - | - |
| | ODTrack-B$_{384}$Zheng et al. (2024) | AAAI24 | 60.9 | - | - | 73.2 | 83.2 | 80.6 | - | - | - |
| | LoRAT-B$_{378}$Lin et al. (2024) | ECCV24 | 59.9 | - | 63.7 | 72.9 | 81.9 | 79.1 | - | - | - |
| Visual-Language trackers | SNLTFeng et al. (2021) | CVPR21 | 27.6 | - | 41.9 | 54.0 | 63.6 | - | 66.6 | - | 80.4 |
| | VLTGuo et al. (2022) | NIPS22 | 53.1 | - | 53.3 | 67.3 | - | 72.1 | 65.3 | - | 85.6 |
| | JointNLT$_{320}$Zhou et al. (2023) | CVPR23 | 56.9 | 69.4 | 58.1 | 60.4 | 73.5 | 63.6 | 65.3 | - | 85.6 |
| | MMTrack$_{384}$Zheng et al. (2023) | TCSVT23 | 58.6 | 75.2 | 59.4 | 70.0 | 82.3 | 75.7 | 70.5 | - | 91.8 |
| | CiteTracker$_{384}$Li et al. (2023) | ICCV23 | 57.7 | 73.6 | 59.6 | 69.7 | 78.6 | 75.7 | 69.6 | 92.2 | 85.1 |
| | UVLTrack-B$_{256}$Ma et al. (2024) | AAAI24 | 63.1 | - | 66.7 | 69.4 | - | 74.9 | 69.3 | - | 89.9 |
| | QueryNLTShao et al. (2024) | CVPR24 | 57.8 | 75.6 | 58.7 | 59.9 | 69.6 | 63.5 | 66.7 | 82.4 | 88.2 |
| | ATSTrack-ViT$_{256}$ | Ours | 64.7 | **83.0** | 68.9 | 70.1 | 80.7 | 76.2 | **71.3** | **87.8** | **95.7** |
| | ATSTrack-ViT$_{384}$ | Ours | **66.2** | **84.2** | **71.5** | **72.6** | **82.4** | **79.5** | 71.0 | 87.6 | 94.4 |
| | Trackers with more advanced backbone | | | | | | | | | | |
| | SUTrack$_{224}$Chen et al. (2025) | AAAI25 | 65.0 | - | 67.9 | - | - | - | 70.8 | - | 93.4 |
| | SUTrack$_{384}$Chen et al. (2025) | AAAI25 | 65.6 | - | 69.3 | - | - | - | 69.7 | - | 91.2 |
| | DUTrack$_{256}$Li et al. (2025) | CVPR25 | 64.9 | 82.9 | 70.6 | 73.0 | 83.8 | 81.1 | 70.9 | - | 93.9 |
| | DUTrack$_{384}$Li et al. (2025) | CVPR25 | 65.6 | 83.2 | 71.9 | **74.1** | **84.9** | **82.6** | 71.3 | - | **95.7** |
| | ATSTrack-HiViT$_{256}$ | Ours | **65.8** | **84.1** | 70.9 | 71.7 | 82.6 | 79.3 | 70.5 | 85.6 | 93.8 |
| | ATSTrack-HiViT$_{384}$ | Ours | **66.8** | **84.5** | **72.7** | 73.4 | 83.9 | 81.4 | **72.1** | **87.9** | 95.2 |

Table 1: Comparison with state-of-the-art visual and visual-language trackers on TNL2K, LaSOT and OTB$_{lang}$. The best two results in each part are shown in **bold** and **bold** respectively.

# 4 EXPERIMENT

## 4.1 IMPLEMENTATION DETAILS

The proposed model is implemented in Pytorch. The models are trained and tested on 4 NVIDIA A6000 GPUs. We utilize CLIP-B as the language backbone and train two versions of models with different visual backbones: ATSTrack-ViT employs vanilla ViT-Base Dosovitskiy (2020) and was trained for 300 epochs with an initial learning rate of $1 \times 10^4$. ATSTrack-HiViT utilizes HiViT-Base Zhang et al. (2023), which is commonly used by latest trackers, and was trained for 200 epochs with an initial learning rate of $2 \times 10^4$. Both versions of the model are optimized using AdamW, with a batch size of 8 and 6,000 samples per epoch. We present the model's speed and number of parameters in A.7.

Our training datasets comprise TNL2K Wang et al. (2021), LaSOT Fan et al. (2019) GOT-10k Huang et al. (2021) and TrackingNet Muller et al. (2018), with an equal sampling ratio. TNL2K and LaSOT contain manually annotated language descriptions, and we use LLM to segment the language descriptions into different attributes. GOT-10k includes annotations for category and motion, we set other attributes to "None". TrackingNet contains category labels, and we use the pre-trained CLIP in Citetracker Li et al. (2023) to predict the color of each target as appearance description.

## 4.2 STATE-OF-THE-ART COMPARISON

We compare our tracker with both state-of-the-art visual and visual-language methods on three commonly used datasets with language annotation, including TNL2K, LaSOT, and OTB$_{lang}$. Results are shown in Table 1.

**TNL2k** contains a total of 2k sequences and 663 words. It introduces two new challenges, i.e. adversarial samples and camera switching, while providing more detailed descriptions, making itself a benchmark specifically dedicated to the visual-language tracking. Our method demonstrates substantial performance enhancement on the TNL2k, with improvements of 3.1% and 1.2% in terms of AUC compared to ViT-based and HiViT-based visual-language trackers respectively. The favorable performance demonstrates the promising potential of our tracker to deal with adversarial samples and modality switch problems.

**LaSOT** is a large-scale long-term tracking benchmark with an average video length of more than 2,500 frames. It includes 1120 sequences for training and 280 sequences for testing. ATSTrack

| Method | AUC | $P_{norm}$ | P |
|---|---|---|---|
| Baseline | 70.6 | 80.7 | 77.1 |
| w/o ASM | 71.1 | 80.6 | 77.4 |
| w/o VFM | 71.6 | 81.5 | 78.4 |
| w/o LFA | 71.5 | 81.2 | 78.4 |
| w/ FGM | **72.0** | **82.1** | **79.0** |

(a) Ablation study of the ASM.

| Method | AUC | $P_{norm}$ | P |
|---|---|---|---|
| w/o token | 72.0 | 82.1 | 79.0 |
| w/o V token | 71.7 | 82.0 | 78.6 |
| w/o L token | 72.0 | 82.4 | 78.6 |
| Attn | 72.4 | **82.6** | 78.9 |
| Concat | **72.6** | 82.4 | **79.5** |

(b) Ablation study of VL token.

| Attr | AUC | $P_{norm}$ | P |
|---|---|---|---|
| w/o Cate | 72.3 | 82.2 | 78.7 |
| w/o App | 72.1 | 81.8 | 78.8 |
| w/o Act | 72.5 | 82.1 | 79.2 |
| w/o Loc | 72.4 | **82.6** | 79.1 |
| Full | **72.6** | 82.4 | **79.5** |

(c) Ablation study of attributes.

Table 2: Ablation Studies of modules in ATSTrack. The best result are shown in **bold**

outperforms the second best ViT-based tracker by 1.8% in terms of AUC, and achieves a performance comparable to SoTA HiViT-based trackers. It could be observed that most Visual-Language trackers perform worse than visual trackers due to the quality of language descriptions Sun et al. (2024b). ATSTrack further narrows this gap with visual trackers, demonstrating that our proposed strategy enables more effective feature refinement. Furthermore, Figure 7 shows detailed results on different attributes in LaSOT.

**OTB**$_{lang}$ Feng et al. (2021) is OTB-100 Wu et al. (2015) dataset extended with a language description of the target object per sequence. It encompasses 11 challenging interference attributes, such as motion blur, scale variation, occlusion, and background clutter. ATSTrack also achieves the state-of-the-art performance with an AUC improvement of 0.8% compared with both ViT-based and HiViT-based trackers.

### 4.3 ABLATION STUDIES

We conduct ablation studies on the LaSOT dataset to verify the effectiveness of each component in our model.

**Effect of Attribute-Specific Modulation.** The ablation results of ASM are shown in Table 2a. We construct a **baseline** by removing components related to language and token propagation mechanism from our model, while preserving template sequence. Performing coarse-grained interaction between language features and visual features through cross-attention (**w/o ASM**) leads to an increase in the AUC by 0.5%. **w/ ASM** shows that the use of attribute-specific modulation improved the AUC score by 0.9% compared to coarse-grained interaction, demonstrating the necessity of reducing the effect caused by the temporal and spatial difference between modality. We also verify the effectiveness of VFM and LFA module by replacing them with cross attention. The results show that **VFM** improves the AUC score by 0.4%, and the **LFA** improves the AUC score by 0.5%. t should be noted that, since not every language description contains all attributes, the actual effectiveness of VFM and LFA is expected to be even higher. Detailed analysis can be found in A.9.

**Effect of Visual-Language token.** The ablation results of VL token are shown in Table 2b. Without the VL token (**w/o token**), the model decreases in the AUC score by 0.6%. We further analyze the influence of information from different modalities. Using **the visual token independently (w/o L token)** does not lead to notable improvements. Using **the Language token independently (w/o V token)** leads to a decrease in the AUC score by 0.3%. These results show that both global visual and language tokens are essential to help the model better understand the target features. We compare different ways to aggregate visual and language information. We have found that performing cross attention between tokens slightly improves the precision but leads to AUC decrease compared to concatenation and chose to concatenate visual and language tokens to acquire VL token.

**Effect of Each attribute.** An important issue in visual-language tracking lies in determining which kinds of description are the most conducive to effective tracking. As shown in Table 2c, removing **category descriptions (w/o Cate)** leads to a decrease in AUC by 0.3%. Removing **appearance descriptions (w/o App)** causes a notable decrease in AUC by 0.5%, as appearance is usually the most obvious factor distinguishing the target from other objects. It should be noted that since existing datasets provide fewer appearance descriptions compared with other attributes, its actual effect would be greater. **The action descriptions (w/o act)** have the weakest impact on tracking results. We consider the reason that action is only useful to distinguishing targets from other similar objects. However, similar objects always share the same actions in existing datasets. **Location descriptions (w/o loc)** also have a weak effect on tracking result. Consider that the location features are already modified by the LFA module, we believe existing location descriptions are more likely to cause interference rather than enhance tracking.

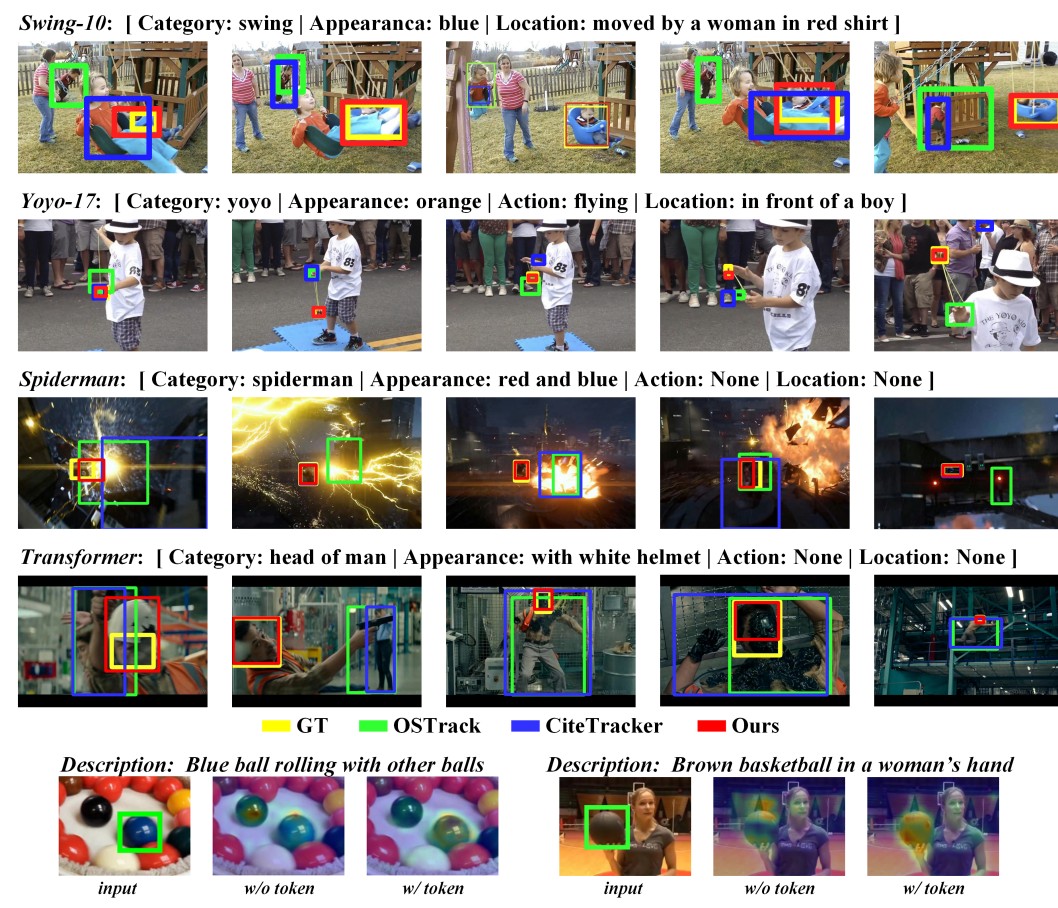

Figure 4: Top half: Visualized results of ATSTrack on challenging scenarios. Bottom half: Visualized results of the VL token

## 4.4 VISUALIZATION

To intuitively demonstrate the excellent performance of the proposed method, we visualize the tracking results of our model and two representative trackers: a visual tracker OSTrack Ye et al. (2022), whose structure closely matches our baseline, and a visual-language tracker CiteTracker Li et al. (2023). In Figure 4, the challenge of performing visual tracking on these four sequences arises from severe occlusion (Swing, Spiderman), fast motion (Yoyo, Spiderman), and view changes (Transform). The results show that ATSTrack outperforms other trackers in these three scenarios, indicating its ability to fully utilize advanced semantic information contained in language descriptions.

Furthermore, we visualize the change of attention maps after introducing the Visual-Language token. As shown in the bottom half of Figure 4, in the ball sequence, the visual backbone pays more attention to the target than distracting object (black ball). In the basketball sequence, the model pays more attention to elements referenced in the language description (basketball and woman) and reduces the focus on irrelevant texture in the background.

## 4.5 CONCLUSION

In this work, we present ATSTrack, which enhances the effect of visual-language tracking by reducing interference caused by the difference in scale of information between visual and language inputs. Specifically, we segment language descriptions into different attributes based on their temporal and spatial correspondence with visual inputs, and modify their features in an attribute-specific manner. Moreover, we introduce a Visual-Language token that comprises modified linguistic information to guide the model to extract visual features that are more relevant to language description. Experiments show that the proposed method achieves a performance comparable to existing methods.

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

# A  APPENDIX

## A.1  EXPERIMENTS OF HYPERPARAMETERS

We conduct experiments on LaSOT  Fan et al. (2019) with different batch sizes and learning rates, the experimental results are shown in Table  3 and Table  4.  We found that the learning rate of $1 \times 10^{-4}$ and batch size of 8 are the optimal.

We also conducted experiments to examine the impact of different gating weight on the effectiveness of LFA. The results are shown in Table  5, our model achieve the best result with gating weight of 50.

| Lr | AUC | $\mathbf{P}_{norm}$ | P |
|---|---|---|---|
| $4 \times 10^{-4}$ | 71.4 | 81.0 | 78.0 |
| $2 \times 10^{-4}$ | 72.0 | 81.8 | 78.7 |
| $1 \times 10^{-4}$ | 72.6 | 82.4 | 79.5 |
| $5 \times 10^{-5}$ | 70.7 | 80.6 | 77.4 |

Table 3: Comparison of different learning rate with batch size of 8.

| Batch | AUC | $\mathbf{P}_{norm}$ | P |
|---|---|---|---|
| 8 | 72.6 | 82.4 | 79.5 |
| 6 | 72.1 | 81.8 | 78.9 |
| 4 | 71.1 | 81.2 | 77.9 |

Table 4: Comparison of different batch size with learning rate of $1 \times 10^{-4}$.

| $\alpha$ | AUC | $\mathbf{P}_{norm}$ | P |
|---|---|---|---|
| 500 | 71.4 | 81.0 | 77.8 |
| 100 | 71.9 | 81.7 | 78.7 |
| 50 | 72.0 | 82.1 | 79.0 |
| 25 | 71.3 | 81.2 | 78.0 |

Table 5: Comparison of different gating weight in LFA.

## A.2  VISUALIZATION ABOUT THE MISALIGNMENT OF LANGUAGE DESCRIPTIONS

The figure provides more examples of misalignment between language and visual inputs, with the misaligned components in the language descriptions highlighted in red.

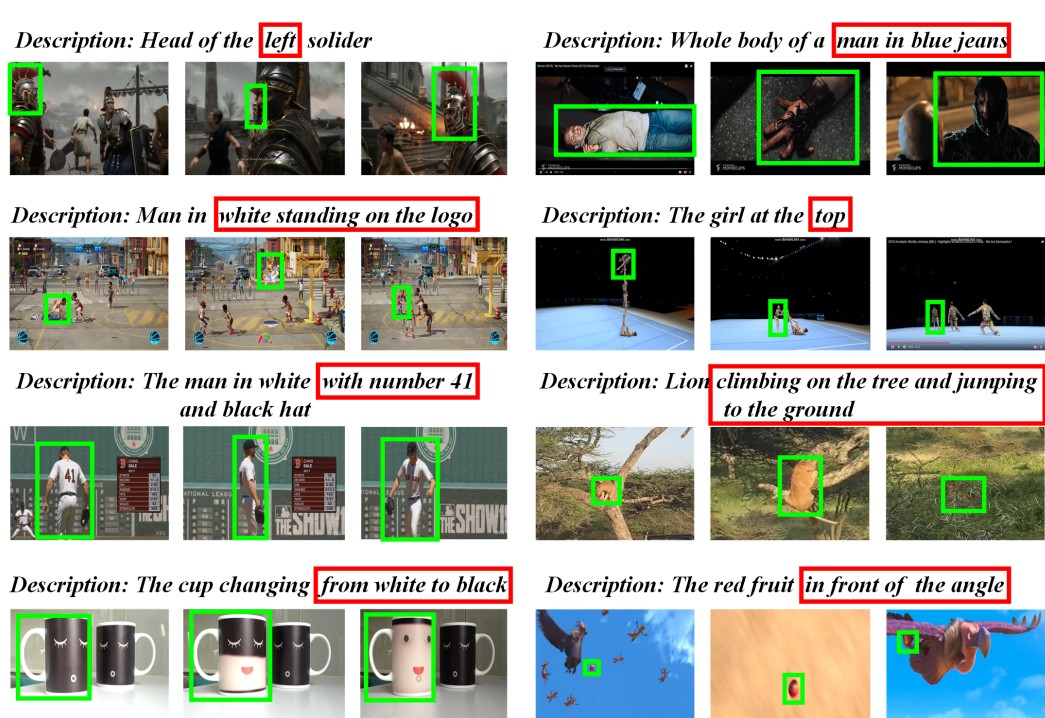

Figure 5: Visualization about the misalignment of language descriptions.

## A.3  THE DESIGN RATIONALE OF ATSTRACK

**The overall model.** Recent studies have begun to recognize the problem of misalignment of language descriptions Li et al. (2023); Shao et al. (2024). However, these works primarily focus on preserving the components of language features that are related to target appearance. For humans,

other components of language descriptions (such as the target's action or its position within the scene) are equally useful for localization. As illustrated in Fig 4, existing tracking models are capable of attending to different instances in the language descriptions (i.e., basketball and woman), while the attention visualization in Chen et al. (2023b) demonstrates that models can capture spatial cues such as the upper-left or lower-right corners of the target. This evidence suggests that current tracking models have the potential to exploit additional components of language descriptions for effective tracking. Motivated by this, our objective is to design a model that can **fully leverages all components of language descriptions**. Once this design goal is established, it naturally leads to the idea of decomposing language into attributes and notice their differences temporal and spatial scales.

**Definition of Attributes.** As emphasized in the main text, we divide the complete language description into four attributes: Category, Appearance, Action, and Location, and provide additional definitions for Action and Location. Beyond separating components with different temporal and spatial scales, this definition of attributes offers the following advantages: 1) For all existing datasets, these attributes can cover nearly all words in each language annotations, leaving very few omissions; 2) This categorization aligns with human linguistic habits, allowing the attributes to be concatenated in the order of Category to Location to form coherent sentences, thereby preserving the contextual relationships of the original descriptions.

**VFM and LFA.** We designed VFM and LFA in a spirit of inspiration by the ideal of feature modification in Shao et al. (2024), and adapted them to better suit the needs of our task. Previous works often bluntly remove background tokens using operations such as top-k selection or binary masking Shao et al. (2024); Chen et al. (2025). This approach is reasonable when the template is only used for visual tracking. However, in our work, we use the template to modulate the target's appearance features, and appearance descriptions may include objects that are not of the same category as the target but are visually distinct due to their shape or characteristics (e.g., "with his sword," "holding an umbrella," "carrying a bag"). To better modulate the appearance features, we hope these objects to be preserved in the template features. Therefore, VFM ultimately adjusts the weight of each token rather than directly removing tokens that do not belong to the target. In LFA, we aim to eliminate misaligned components while preserving as much valid information as possible. From our observations, a natural conclusion emerges: more general and broad descriptions (containing less valid information and with more consistent information across tokens) are usually more accurate, whereas more detailed descriptions, due to including more specifics (more information and varying emphasis across tokens), are naturally more prone to misalignment with the target's state. Based on this insight, we design a dynamic threshold that reflects the consistency among tokens. In addition, we employ a sigmoid function to project tokens with low similarity to visual features close to zero, achieving stronger elimination.. As shown in Table 2a, both VFM and LFA are more effective than standard cross-attention.

## A.4 DETAILS ABOUT THE USEGE OF LLM

We use Kimi Chat Team et al. (2025), a Large Language Model (LLM) equipped with exceptional contextual understanding ability produced by Moonshot AI, to segment the language description into four attributes. The prompt we use is as follows:

*Each line in the following document is a description of a specific target. Please divide each description into the following four parts: "Category", "Appearance", "Action", and "Location". The general characteristics of each section are as follows: "Category" should be the subject of this sentence; "Appearance" is a description of the appearance of the subject; "Action" usually refers to the action of the subject itself or interaction with accessory items such as handheld items, backpacks, etc.; "Location" includes the position of the subject and the interaction between the subject and other independent objects, such as playing tennis together or chasing each other. Here is an example of segmentation. Please strictly follow this format for output: "Yellow airplane flying in the air" will be segmented into "Category:airplane\n Color:yellow\n Action:flying\n Location:in the air\n".*

It should be noted that the segmentation results acquired with LLM still require manual adjustment.

## A.5 EFFECT OF MUTIPLE TEMPLATES

Recently, many single-object trackers have begun to incorporate multiple templates and achieve improved performance by providing richer temporal context. To achieve temporal scale alignment, we also adopt multiple templates. For a fair comparison, we conduct experiments with different numbers of templates, and the results are shown in Table 6 and Table 7. Together with Table 1, it can be observed that ATSTrack$_{ViT}$ surpasses all ViT-based trackers even with single template. ATSTrack$_{HiViT}$ achieves comparable performance to HiViT-based trackers using three (DUTrack Li et al. (2025)) and two templates (SUTrack Chen et al. (2025)), respectively.

We uniformly sampled templates from historical tracking results during inference. Our model achieved optimal results with three templates, which is consistent with that of ODTrack Zheng et al. (2024). By observation, we found that excessively long template sequences increase the probability of capturing templates with occluded targets or tracking errors, thereby degrading tracking performance.

| Num | LaSOT | | | TNL2K | | |
|---|---|---|---|---|---|---|
| | AUC | $P_{norm}$ | P | AUC | $P_{norm}$ | P |
| 1 | 71.7 | 81.8 | 78.2 | 64.7 | 81.7 | 69.2 |
| 2 | 72.1 | 82.0 | 78.8 | 65.5 | 83.1 | 70.3 |
| 3 | **72.6** | 82.4 | **79.5** | **66.2** | **84.2** | **71.5** |
| 4 | 71.8 | **82.5** | 78.0 | 64.2 | 81.5 | 69.3 |
| 5 | 71.0 | 81.2 | 77.6 | 63.9 | 81.2 | 69.3 |

Table 6: ATSTrack$_{ViT}$ with different number of templates.

| Num | LaSOT | | | TNL2K | | |
|---|---|---|---|---|---|---|
| | AUC | $P_{norm}$ | P | AUC | $P_{norm}$ | P |
| 1 | 72.0 | 82.0 | 79.0 | 65.3 | 83.4 | 70.2 |
| 2 | 72.8 | 82.9 | 80.1 | 66.0 | 84.2 | 71.1 |
| 3 | **73.4** | **83.9** | **81.4** | **66.8** | **84.5** | **72.7** |
| 4 | 72.6 | 82.3 | 79.4 | 66.1 | 83.9 | 71.1 |
| 5 | 71.9 | 81.9 | 78.9 | 64.9 | 82.0 | 69.7 |

Table 7: ATSTrack$_{HiViT}$ with different number of templates.

## A.6 MAPPING FUNCTION IN VFM

For an input $X$ of size $L \times C$, we first apply a linear mapping along each channel to project it into the range $[0, 1]$, allowing us to determine the importance of each token at the channel level.

$$X_j^{\text{norm}} = \frac{X_j - \min(X_j)}{\max(X_j) - \min(X_j) + \epsilon}, \quad j = 1, \ldots, C \tag{12}$$

where $j$ represents the $j$-th channel. For $X^{\text{norm}}$, we compute its average value along the channel dimension to obtain an importance score for each token $X^{\text{score}} \in [0, 1]^{L \times 1}$, and then use a mapping function to project the importance scores into the range $[\alpha, 1]$ to obtain the final weight matrix.

$$X_i^{\text{score}} = \frac{1}{C} \sum_{j=1}^{C} X_{i,j}^{\text{norm}} \tag{13}$$

$$W = (1 - \alpha) \cdot \frac{X^{\text{score}} - \min X^{\text{score}}}{\max X^{\text{score}} - \min X^{\text{score}} + \epsilon} + \alpha \tag{14}$$

Through the above operations, we obtain a token-level weight matrix. The parameter $\alpha$ ensures that semantically relevant attachments are not completely eliminated, and we set $\alpha$ to 0.2.

## A.7 SPEED AND PARAMETERS

Table 8 shows the number of parameters and FLOPs of our ATSTrack. Our model achieves an average speed of 49fps on the LaSOT dataset, achieving real-time performance and comparable with other advanced models. We avoid additional computational overhead from using CLIP Radford et al. (2021) by extracting language features only in the initial frame and saving them for subsequent inference.

## A.8 USING BERT AS LANGUAGE BACKBONE

Existing vision-language trackers either employ CLIP Radford et al. (2021) or RoBERTa Liu et al. (2019) as the language backbone. Our model utilizes CLIP. For a more comprehensive presentation,

| Method | Params | Flops | Speed | Device |
|---|---|---|---|---|
| ATSTrack$_{HiViT384}$ | 77M | 71G | 40 fps | 3090 |
| ATSTrack$_{ViT384}$ | 93M | 90G | 49 fps | 3090 |
| ODTrackZheng et al. (2024) | 92M | 73G | 32 fps | 2080Ti |
| SeqTrackChen et al. (2023b) | 89M | 148G | 11 fps | 2080Ti |
| ARTrackWei et al. (2023) | 181M | 172M | 13.5 fps | V100 |

Table 8: Comparison of parameters and speed.

we trained a version that uses RoBERTa as the language backbone, and the results are shown in the 9. It should be noted that since RoBERTa produces variable-length outputs, and our model involves direct matrix addition, which requires structural modifications (adaptive interpolation) to accommodate variable-length features, the results in 9 are mostly for reference and could not precisely reflect the upper limit of our model's capabilities.

| Method | LaSOT | | | TNL2K | | |
|---|---|---|---|---|---|---|
| | AUC | P$_{norm}$ | P | AUC | P$_{norm}$ | P |
| CLIP | 73.4 | 83.9 | 81.4 | 66.8 | 84.5 | 72.7 |
| RoBERTa | 72.9 | 83.0 | 80.3 | 66.0 | 84.1 | 71.7 |

Table 9: Comparison of different language backbone.

## A.9 QUANTITATIVE DIFFERENCES OF ATTRIBUTES

We divide the complete language description of the target into four phrases with different attributes to align the temporal and spatial scales between different visual and language inputs. We conduct quantitative analysis on different attributes on TNL2KWang et al. (2021) and LaSOTFan et al. (2019), which play important roles in both the training and testing sets. As shown in 6, the number of **Appearance** descriptions is less than other attributes, which means that the influence of appearance descriptions on tracking results may be greater than that reflected in the ablation experiment. The average length of **Location** is significantly larger than other attributes, suggesting to a certain degree that location descriptions are more likely to contain information that does not align with visual input.

## A.10 ANALYSIS OF FAILURE CASES

We have already demonstrated the improvements of ATSTrack over other models through visualizations. This section mainly focuses on intuitively analyzing the scenarios in which our modifications lead to degradation compared with the baseline through visualization.

**Ambiguity of the language description.** The first two sequences in Fig 8 illustrate the cases where our model degrades due to ambiguities in the language description. Specifically, when multiple distractors in the scene exhibit extremely similar appearances to the target, we expect the language description to help the model differentiate them. However, in these two cases, the language description itself matches the distractors and lacks discriminability, which increasing the likelihood of distraction and making recovery from tracking errors more difficult.

**Interference from secondary components.** demonstrate that the model may be disturbed by other components in the sentence (i.e. panda, boy). It is worth noting that such cases are rare, typically occurring only when the distractor overlaps with the target itself, and the model usually corrects itself quickly. Nevertheless, this phenomenon also reveals that it could be a potential direction for improving vision-language tracking by providing the model with clearer cues to distinguish foreground from background Sun et al. (2024b).

## A.11 LIMITATION AND FUTURE WORKS

In this paper, we have achieved significant improvements in feature modification by minimizing the interference caused by the temporal and spatial discrepancies between visual and language inputs. Consequently, we have mitigated the adverse effects of mismatches between visual and language

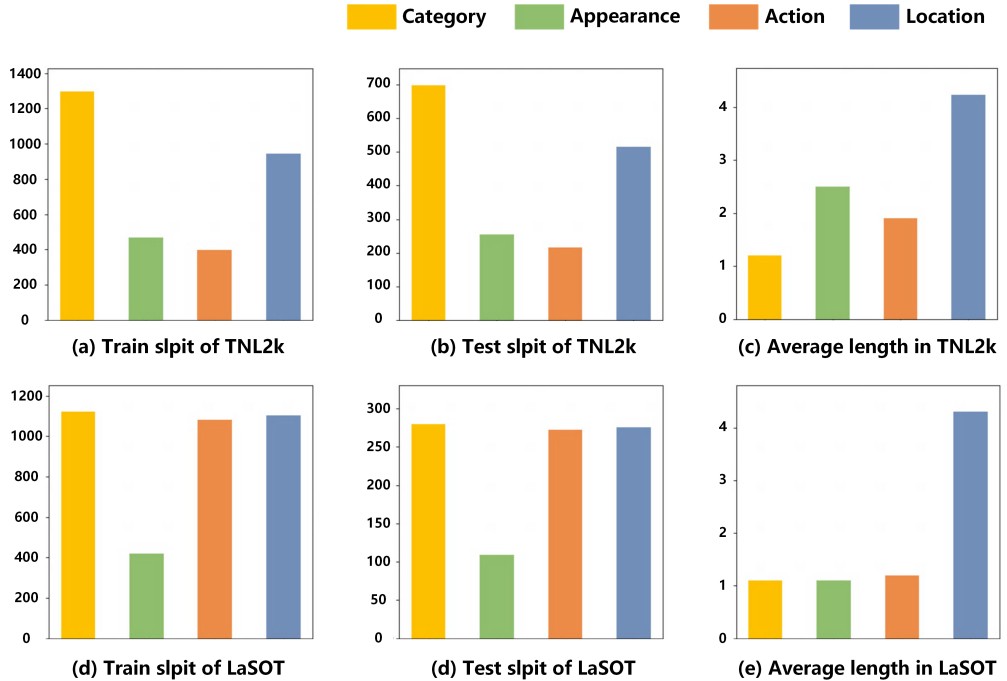

Figure 6: Quantitative analysis on different attributes on TNL2K and LaSOT dataset.

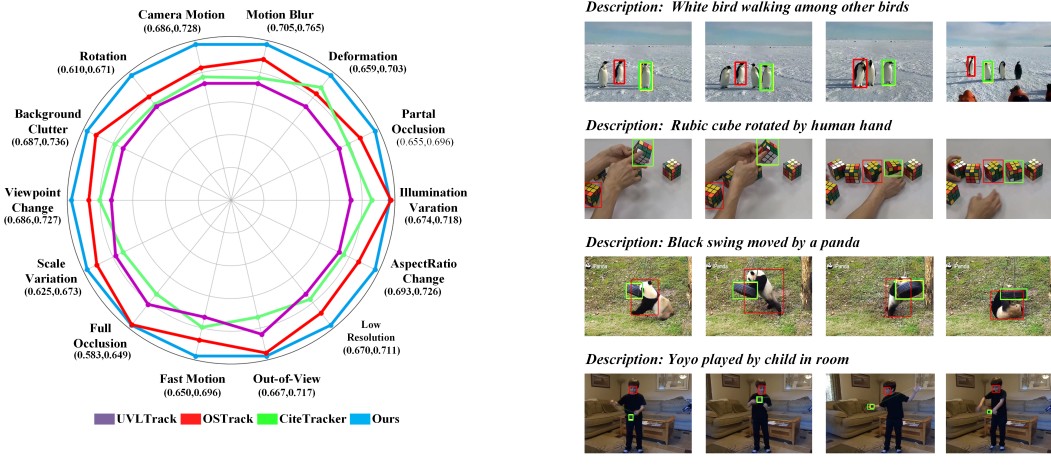

Figure 7: AUC scores of different attributes in LaSOT.

Figure 8: Visualization of failure cases

inputs on visual-language tracking. We believe that our proposed decomposition of inputs based on temporal and spatial scales could be an important approach to enhancing the cross-modal understanding capability of trackers, and we will continue to pursue this direction. However, our work has not yet fully addressed the underlying problem as we did not fundamentally alter the information the model receives, and it remains constrained by the content and style of manually annotated language descriptions. Currently, some works are exploring the generation of textual descriptions through multi-modal large models Sun et al. (2024b); Li et al. (2025). However, this approach requires substantial computational resources and does not align with the fundamental requirements of tracking. In our future work, we will focus on exploring lightweight methods for updating language features.

## A.12 MORE VISUALIZATION RESULTS

To demonstrate the advantages of our proposed ATSTrack more intuitively, we provide more visualization results compared with OSTrack Ye et al. (2022), and CiteTrackerLi et al. (2023) in 9. Results demonstrate that ATSTrack achieves favorable performance in a variety of challenging scenarios including fast motion, severe occlusion, similar targets and deformation.

*Bottle-12*: [ Category: bottle | Appearanca: None | Action: hanging | Location: in the air ]

*Zebra-16*: [ Category: zebra | Appearance: None | Action: running | Location: on dry grass with other zebras ]

*Train-1*: [ Category：train | Appearance: blue | Action: moving | Location: on its track ]

*Dog-19*: [ Category: dog | Appearanca: large | Action: None | Location: leading a group of dogs ]

*Robot-19*: [ Category: robot | Appearance: grey | Action: fighting with other robots | Location: on white table ]

*Yellowman*: [ Category: yellowman | Appearance: yellow | Action: looking at the right | Location: None ]

*Ironman*: [ Category: head of man | Appearance: in blue | Action: None | Location: None ]

GT    OSTrack    CiteTracker    Ours

Figure 9: Visualized results of the proposed ATSTrack.

