# OpenReview forum: "ATSTrack: Enhancing Visual-Language Tracking by Aligning Temporal and Spatial Scales"
_ICLR.cc/2026/Conference — ICLR 2026 Conference Withdrawn Submission_

### Official Review · Reviewer_5VDT · 2025-10-17

**Soundness:** 2
**Presentation:** 1
**Contribution:** 2
**Rating:** 4
**Confidence:** 5

**Summary:**

Theis work proposes ATSTrack, a method that aims to improve the alignment between textual and visual modalities by addressing the scale discrepancy between them. They design an attribute-specific modification module to alleviate temporal and spatial misalignment between features, and further introduce a visual-language token to enhance cross-modal correlation and guide visual feature extraction.

**Strengths:**

1. The issue addressed in this paper, namely the misalignment between text prompts and dynamic visual targets, is a core problem in the visual-language tracking task.

2. This paper includes many illustrative figures, which facilitate the reader’s quick understanding.

**Weaknesses:**

1. The motivation of this paper, which is to align textual cues with dynamic video features, shares many similarities with a recent visual-language tracker, ATCTrack. However, this paper lacks discussion and comparison with ATCTrack (ICCV 2025).

2. A key design of this paper is to divide the text prompt into four parts. What is the basis for this division? Given the flexible and diverse forms of text descriptions, how can this division ensure coverage of all types of texts?

3. The baseline models compared in this paper do not include ATCTrack. On the TNL2K and LaSOT benchmarks, the performance of the proposed method is evidently weaker than that of ATCTrack . Therefore, the claim of "outperforms state-of-the-art vision-language trackers" is not convincing.

4. The method proposed in this paper involves complex text processing. I believe it should be evaluated on the recently proposed benchmark MGIT, which contains fine-grained and detailed text annotation information.

5. There are issues with the citation of references in this paper. Specifically, the cited papers lack parentheses for differentiation, which makes reading somewhat inconvenient.

6. This paper employs LLM for text component analysis but does not provide an explanation of the use of LLM in the manuscript (which is a requirement for this paper submission).

**Questions:**

Please refer the above Weaknesses.

---

> ### Author Response · Authors · 2025-11-12
> **Response to Reviewer 5VDT**
>
> First of all, we sincerely thank you for your professional and insightful comments. We truly appreciate your deep expertise in the field of single-object tracking. Even though we did not have high expectations for acceptance, we genuinely value the opportunity to engage in this discussion with you. Below, we provide detailed responses to your comments.
>
> ---
>
> ### **1. Difference from ATCTrack**
>
> In fact, our work and **ATCTrack** were developed during the same period, and our manuscript was even made publicly available slightly earlier. Although there are indeed some superficial similarities between **ATSTrack** and **ATCTrack** (both segment sentences and have similar names), this is purely coincidental (and unfortunate for us). Importantly, the motivations of the two methods are fundamentally different.
>
> **ATCTrack** emphasizes the linguistic components most relevant to the target itself, training a model to identify these key words and align their semantics with visual-temporal cues. In contrast, our method seeks to fully exploit *all* components within the sentence. Specifically, we observe that different parts of a sentence convey information at distinct temporal and spatial scales, and based on this observation, we design an **attribute-specific method** to refine the features accordingly.
>
> It would be unfair to directly compare our approach with **ATCTrack**, since it benefits from a stronger architecture (**AQATrack** vs. **ODTrack**), better initialization weights (**FastITPN** vs. original **HiViT**), larger training data (8 datasets vs. 4), and more instances per epoch (10k vs. 6k). As shown in **ATCTrack**’s original paper, the baseline of their method already demonstrates highly competitive performance. In contrast, our method outperforms all approaches built upon similar architectures and experimental settings. Moreover, our ablation studies clearly demonstrate the effectiveness of each proposed component, which we believe sufficiently validates the value of our approach.
>
> ---
>
> ### **2. Rationale for Attribute Selection**
>
> We chose the four attributes based on empirical observation: for any sentence that describes a specific target at a given time, these attributes are sufficient to capture all relevant content. We also consulted ChatGPT, which could not provide a counterexample.
>
> As mentioned in Section 3.2 of the main paper, our definitions were designed to cover as many components as possible. In practice, this attribute design allows us to segment all language annotations in our dataset **without omission**.
>
> ---
>
> ### **3. Experiments on MGIT**
>
> MGIT  is an excellent work, and we plan to include experiments on it. However, because MGIT contains multi-granularity rather than fine-grained annotations, we need to perform re-annotation before training. Furthermore, our GPUs are currently occupied, so these experiments could take a bit longer to complete.
>
> ---
>
> ### **4. Clarification on the Use of LLMs**
>
> We have provided a detailed explanation in **Appendix A.4**.
>
> ---
>
> Finally, we would like to thank you again for pointing out the formatting issues in our manuscript. We will carefully correct them in the revised version.

---

> > ### Comment · Reviewer_5VDT · 2025-11-26
> >
> > Thank you for your response. However, several issues remain unresolved:
> >
> > **Regarding Q1 (Difference from ATCTrack):**
> > You mentioned that ATCTrack incorporates certain practical techniques that contribute to its improved performance. A more convincing and rigorous approach would be to integrate these techniques into your own method and demonstrate that ATSTrack can outperform ATCTrack under comparable settings.
> >
> > **Regarding Q2 (Rationale for Attribute Selection):**
> > Stating that the choice is based solely on “empirical observation” is not sufficiently convincing. Relevant literature, theoretical support, or prior studies should be provided to justify the selection of these attributes.
> >
> > **Regarding Q3 (Experiments on MGIT):**
> > Do you now have evaluation results on this benchmark?
> >
> > Considering the above issues and the comments from other reviewers, I will temporarily maintain my current rating.

---

### Official Review · Reviewer_Jibo · 2025-10-22

**Soundness:** 2
**Presentation:** 3
**Contribution:** 2
**Rating:** 2
**Confidence:** 4

**Summary:**

To address the spatio-temporal scale mismatch between visual information and linguistic descriptions in visual-language tracking, this paper proposes ATSTrack, which enhances the effectiveness of feature modification by Aligning the Temporal and Spatial (ATS) scales of different input components.

Specifically, temporal alignment is achieved by segmenting and modifying the language modality to align with the visual information, and by creating a token that fuses visual and linguistic information.

**Strengths:**

The methodology is described in detail, and the accompanying figures clearly and accurately convey the authors' intent.

The model design possesses notable interpretability: by leveraging a Large Language Model (LLM) to segment attributes within the linguistic information, it facilitates the analysis of the roles played by different attributes in the tracking task.

The experimental evaluation is comprehensive, encompassing experiments conducted with both HiViT and ViT serving as the backbone architectures.

**Weaknesses:**

**Lack of Novelty**: The issue of spatio-temporal scale mismatch has already been noted and partially addressed by multiple preceding studies in visual-language tracking. The paper fails to discuss the distinctions between its approach and these existing works. Furthermore, the proposed method appears to be primarily a synthesis of existing literature with minor modifications [1][2].

**Prohibitive Overhead**: The utilization of a Large Language Model (LLM) for natural language processing incurs excessive computational costs, rendering a fair comparison with other trackers.

**Failure to Achieve SOTA**: The method does not achieve state-of-the-art (SOTA) performance; specifically, it underperforms DUTrack[1] on the LaSOT benchmark. Moreover, the best-performing result in the final column of Table 1 is incorrectly marked.

[1] Li, Xiaohai, et al. "Dynamic Updates for Language Adaptation in Visual-Language Tracking." Proceedings of the Computer Vision and Pattern Recognition Conference. 2025.
[2] Feng, Xiaokun, et al. "ATCTrack: Aligning Target-Context Cues with Dynamic Target States for Robust Vision-Language Tracking." arXiv preprint arXiv:2507.19875 (2025).

**Questions:**

Regarding the use of a single token to convey spatio-temporal visual-language information, its sufficiency is questionable. It is recommended to supplement the study with ablation experiments on the number of tokens, along with visualizations of the attention maps corresponding to this token.

What is the justification for the manual segmentation of attributes (i.e., Category, Appearance, Action, Location)? Does this methodology risk overlooking the intrinsic correlations among these attributes? Could alternative, more effective processing strategies exist?

---

> ### Author Response · Authors · 2025-11-14
> **Response to Reviewer Jibo**
>
> We greatly appreciate your efforts and provide the following responses:
>
> ## 1. Innovation Statement
> In fact, our work and ATCTrack were developed concurrently, and our manuscript was even made publicly available **slightly earlier**. Although there are indeed some superficial similarities between ATSTrack and ATCTrack (both segment sentences and share similar names), this is purely coincidental. Importantly, the **motivations of the two methods are fundamentally different.**
>
> ATCTrack emphasizes the **linguistic components most relevant to the target itself**, training a model to identify these key words and align their semantics with visual-temporal cues. In contrast, our method **seeks to fully exploit all components**  within the sentence. Specifically, we observe that different parts of a sentence convey information at distinct temporal and spatial scales, and based on this observation, we design an attribute-specific method to refine the features accordingly.
>
> DUTrack is **almost entirely unrelated** to our ATSTrack. It utilizes additional cross-modal pre-trained models to **dynamically update language descriptions**, whereas our model  focuses on **refining linguistic features through visual information**.
>
> Moreover, the submodules within the proposed Attribute-Specific Modulation are entirely different from those in DUTrack and ATCTrack, so describing ATSTrack as a simple synthesis of the two is completely unfounded.
>
> ## 2. Impact of Large Models
> ATSTrack employs a large model only **once per video sequence** to segment language descriptions. Therefore, the computational overhead introduced by the large model are minimal.
>
> ## 3. Clarification on "Achieving SOTA"
> First, we have **not claimed**that ATSTrack "achieves SOTA." We merely stated that "ATSTrack achieves performance **comparable to existing methods**." Second, we believe our method remains competitive. Despite DUTrack's  **frequent invocation of large models** and use of **better initial weights (Fast-iTPN vs. original HiViT)**, our model still outperforms DUTrack on TNL2K and OTBlang.
>
> ## 4. VL Tokens
> We acknowledge that extending the VL token propagation mechanism is indeed a potential improvement direction for ATSTrack. We will seriously consider supplementing experiments on token length. The bottom row of **Figure 4** already includes visualizations of attention maps of using VL tokens, which we hope address your questions about VL tokens.
>
> ## 5. Rationale for Attribute Selection
> We selected the four attributes based on empirical observation: for any sentence describing a specific target at a given time, these attributes are sufficient to capture all relevant content. We also consulted ChatGPT, which could not provide a counterexample. As mentioned in Section 3.2 of the main paper, our definitions were designed to cover as many components as possible. In practice, this attribute design allows us to segment all language annotations in our dataset without omission.

---

> ### Author Response · Authors · 2025-11-14
>
> We didn't spot any mistakes in the last column of Table 1. We know that sometimes the paper sent by OpenReview  to reviewers doesn't show underlines correctly, did you run into this issue by any chance? If not, could you let us know what exactly goes wrong?

---

### Official Review · Reviewer_JR2b · 2025-10-29

**Soundness:** 2
**Presentation:** 3
**Contribution:** 2
**Rating:** 4
**Confidence:** 4

**Summary:**

This paper proposes ATSTrack, a spatio-temporal alignment framework for vision-language tracking. The method decomposes language descriptions into attributes (category, appearance, action, location) and employs VFM, LFA, and a cross-frame VL token for fine-grained visual-language alignment and temporal consistency. Experiments on multiple benchmarks show improved performance over existing methods, especially in challenging scenarios.

**Strengths:**

The paper introduces an original perspective on vision-language tracking through attribute-specific spatio-temporal alignment, showing a clear methodological novelty. The approach is well-designed, combining attribute decomposition with cross-frame priors for robust feature fusion. Experiments are comprehensive and results are consistent across benchmarks, demonstrating strong technical quality. The paper is clearly written, with well-defined modules and intuitive structure. Overall, it contributes a valuable direction for improving interpretability and robustness in dynamic vision-language tasks.

**Weaknesses:**

1) The experiments focus mainly on short-text benchmarks such as TNL2K and LaSOT, which do not fully capture the advantages of the proposed framework under long-term, evolving, or distractive scenarios. It is recommended to include MGIT experiments to verify the effectiveness of the cross-frame semantic and spatio-temporal alignment mechanisms.

2) The attribute decomposition relies on external rules or LLM-based parsing rather than a learnable design, which may affect reproducibility and cross-domain generalization. The cross-frame VL prior may drift under scene changes or semantic shifts, yet no failure detection or reset strategy is analyzed.

3) The paper lacks a systematic robustness study under textual noise, ambiguous references, and action phase transitions, as well as missing computational complexity and efficiency reports. Adding these analyses would provide a more comprehensive evaluation of the model’s stability and applicability.

**Questions:**

1) Could the authors include experiments on MGIT or other long-term, multi-granular benchmarks to better demonstrate the advantage of cross-frame semantic alignment?

2) Is the attribute decomposition module learnable? If it relies on external rules or LLM parsing, how does the method ensure consistency and reproducibility across different text styles or domains?

3) How does the cross-frame VL token handle potential drift under scene switches or abrupt semantic changes? Is there any reset or down-weighting mechanism?

4) When language cues (e.g., appearance or location) conflict with visual evidence, how does the model arbitrate or calibrate confidence?

5) How robust is the model to textual noise, ambiguous references, or action phase transitions? Are there quantitative or qualitative analyses?

---

> ### Comment · Reviewer_JR2b · 2025-11-26
>
> Since there is still about a week left for the rebuttal, I wanted to check in again and ensure my questions are clear before the DDL. I noticed that my original comments haven't been replied to yet, but I read through your discussions with the others, and it turns out most of my earlier concerns were already addressed there. So those parts are all fine for me now.
>
> But I still have one main thing I want to understand better: **how your method behaves in long-term, multi-granularity, and dynamically changing scenarios.** For example, in longer sequences like MGIT, the target appearance may slowly change, the scene might shift, and the text description can become less accurate over time. So I'm mainly curious about: 1)how the attribute-specific mechanism adapts over time in these long sequences, 2) how the model balances visual evidence vs. the text cues when they start to mismatch, and 3) whether the VL token drifts over time, and if so, how the model deals with it.
>
> I also noticed some reviewers asked about the relation to ATCTrack. Personally, I think in your rebuttal, mentioning things like ”our manuscript was even made publicly available slightly earlier” isn’t really necessary (not a big problem, just a kind suggestion, because mentioning it may not be ideal during the double-blind policy). What I’m more interested in is the conceptual difference: ATCTrack focuses more on extracting the most target-related language parts, while your method tries to use the whole sentence and lets different parts work at different temporal/spatial scales.
>
> So, the only thing I’m really hoping to hear more about is this “long-sequence / dynamic scenario” question. If it’s easier, feel free to add some example or visualization cases in the updated version, I can check it directly there, which might be even clearer.

---

> ### Author Response · Authors · 2025-11-26
> **Response to Reviewer JR2b**
>
> First of all, we sincerely thank you for your efforts and time. We appreciate the valuable questions and comments you have raised, and we are glad to provide further clarification and discussion. Our responses to your questions are as follows:
> **1. Reliability of Attribute Decoupling**
> The language descriptions of a target at a specific moment are typically not overly complex,  and our proposed attribute decoupling strategy is also straightforward, current LLMs can handle it effectively without significant generalization issues. Our method has already demonstrated a certain degree of generalization to different language styles on existing datasets: it performs well on LaSOT with short descriptions as well as on TNL2K with longer and more detailed descriptions (as shown in Table 6 of the appendix). More complex descriptions, however, go beyond the scope of all existing studies and available datasets.
>
> **2. Ambiguity between Language and Visual Information**
> Indeed, ATSTrack aims to address the failure and ambiguity of language descriptions caused by target motion. Specifically, since visual features always contain accurate information, existing methods typically adjust language features with visual information to filter out invalid information before performing cross-modal fusion. Building on this idea, our method further enhances the adjustment of language features by leveraging temproal and spatial  aligned attribute-level refinement and cross-modal temporal prior propagation.
>
> **3. Scene Changes and Rapid Semantic Variation**
> Scene changes and rapid semantic variations are indeed meaningful challenges in single-object tracking. Given the high-level abstraction of language, language information can retain useful cues better than visual information under scene changes. Therefore, our method theoretically performs better than approaches that rely solely on visual temporal priors in such scenarios. However, in current datasets, these situations account for only a small fraction, making quantitative evaluation difficult even with appropriate optimizations. We will consider investigating target tracking under scene changes as a direction for future work.
>
> **4. Complexity and Efficiency**
> We have provided an analysis of efficiency in Table 8 of  the appendix.
>
> Finally, we commit to conducting experiments on MGIT as soon as possible.

---

> > ### Comment · Reviewer_JR2b · 2025-11-26
> >
> > Thanks for the timely response. I appreciate the clarifications.
> >
> > I still have a small suggestion regarding the style of your rebuttal: at the moment, most of the replies are explanatory in nature, but as a reviewer, I would be more convinced if I could see more substantive evidence. This could be in the form of new experiments (e.g., on MGIT or Lasot-ext, which multiple reviewers mentioned), or a deeper analysis of existing results, such as more fine-grained qualitative discussions or new quantitative cases on longer or more challenging sequences.
> >
> > I also want to share a few minor suggestions that may help further improve the paper (these are just small details, for your reference):
> > - Currently, all table captions are placed below the tables, which is more like AAAI style; for ICLR, it is more standard to put table captions above the tables.
> > - It is better to end each displayed equation with proper punctuation, as the equation is part of the sentence.
> > - In the ICLR template, the more standard way to cite is \citep rather than \cite, so that citations appear in parentheses instead of being directly glued to the main text. Right now, most citations are attached directly to the sentence without brackets.
> > - Regarding the issue another reviewer mentioned for Table 1: it seems that the value in the last row and last column is mislabeled. The best result should be 95.7 rather than the 95.2 currently shown.
> >
> > These are all just suggestions to help with polishing and presentation. I still look forward to seeing more substantial experimental updates from your side in the final week of the rebuttal period.

---

### Official Review · Reviewer_ifMR · 2025-10-29

**Soundness:** 2
**Presentation:** 2
**Contribution:** 2
**Rating:** 2
**Confidence:** 5

**Summary:**

The paper proposes ATSTrack, a visual-language tracker designed to alleviate the temporal and spatial scale misalignment between visual and textual modalities. The method decomposes textual descriptions into four attributes. It introduces two main modules: (1) an Attribute-Specific Modulation (ASM) module to refine cross-modal feature interaction, and (2) a Visual-Language token (VL token) that propagates linguistic context across frames to guide visual feature extraction. Experiments on TNL2K, LaSOT, and OTBlang datasets show competitive results, and several ablation studies are provided to justify the design choices. While the paper is clearly written and the motivation is understandable, the proposed method’s novelty and experimental depth are limited. The improvements over existing approaches are marginal, and the claimed benefits of attribute-level decomposition and scale alignment are not convincingly demonstrated.

**Strengths:**

1. The paper is well structured, and the problem of temporal and spatial misalignment between visual and language modalities is clearly described and intuitively motivated.
2. The method is evaluated on three major visual-language tracking benchmarks (TNL2K, LaSOT, OTB-lang) and shows consistent results.

**Weaknesses:**

1. Insufficient comparison with state-of-the-art methods. Although the paper includes comparisons with several previous works (e.g., CiteTracker, QueryNLT, DUTrack), it omits stronger and more recent SOTA baselines, especially those leveraging multimodal large language models (e.g., ChatTracker[1], and ATCTrack[2]).
2. Insufficient evaluation datasets. More vision-language tracking datasets should be added to evaluate the tracker's performance, including MGIT and LaSOText.
3. Limited novelty. The proposed “temporal and spatial scale alignment” mainly relies on attribute-level decomposition and separate processing, similar to existing vision-language trackers such as ATCTrack. The VL token design is only a lightweight extension of existing cross-frame feature propagation mechanisms with minimal methodological innovation(e.g. AQATrack's query[3], ODTrack's Temporal tokens[4] and MambaVLT's hidden states[5]).
4. Weak validation of the attribute decomposition strategy. The segmentation of attributes depends heavily on an LLM-based prompt (Appendix A.4) and manual correction, but there is no quantitative evaluation of segmentation quality or its sensitivity to errors. The paper claims to “align temporal and spatial scales,” yet provides no quantitative evidence demonstrating that this alignment is achieved. Although the paper presents some visualization results, it offers little theoretical analysis or reasoning to substantiate why the proposed method achieves its improvements.

[1] Sun Y, Yu F, Chen S, et al. Chattracker: Enhancing visual tracking performance via chatting with multimodal large language model[J]. Advances in Neural Information Processing Systems, 2024, 37: 39303-39324.
[2] Feng X, Hu S, Li X, et al. ATCTrack: Aligning Target-Context Cues with Dynamic Target States for Robust Vision-Language Tracking[C]//Proceedings of the IEEE/CVF International Conference on Computer Vision. 2025: 19850-19861.
[3] Xie J, Zhong B, Mo Z, et al. Autoregressive queries for adaptive tracking with spatio-temporal transformers[C]//Proceedings of the IEEE/CVF Conference on Computer Vision and Pattern Recognition. 2024: 19300-19309.
[4] Zheng Y, Zhong B, Liang Q, et al. Odtrack: Online dense temporal token learning for visual tracking[C]//Proceedings of the AAAI conference on artificial intelligence. 2024, 38(7): 7588-7596.
[5] Liu X, Zhou L, Zhou Z, et al. Mambavlt: Time-evolving multimodal state space model for vision-language tracking[C]//Proceedings of the Computer Vision and Pattern Recognition Conference. 2025: 8731-8741.

**Questions:**

1.  How does the ATSTrack perform on datasets such as MGIT and LaSOText? How does ATSTrack perform compared to MambaVLT and ATCTrack on common visual-language tracking benchmarks?
2. How does your VL token differ functionally from existing temporal or memory tokens?
3. Your attribute segmentation relies heavily on an LLM-based prompt (Appendix A.4) and manual correction. How consistent are these annotations, and how sensitive is model performance to errors in this decomposition?
4. Could you provide theoretical reasoning or analytical justification to explain why the proposed attribute decomposition improves tracking performance?

---

> ### Author Response · Authors · 2025-11-20
> **Response to Reviewer ifMR**
>
> First, we sincerely thank you for your thorough and insightful review. We noticed that you are very familiar with vision–language tracking, and we truly appreciate the opportunity to further communicate with an expert in this area. Below are our responses to your comments.
>
> **1. Principle of selecting baseline methods.**
> When comparing with existing methods, our goal is to *minimize the influence of visual information on the final results* by unifying the visual backbones and tracking frameworks as much as possible. This allows us to more clearly highlight each model’s ability to leverage language information.
> For this reason, in the comparison table we did not include several methods that were already discussed elsewhere in the paper, as their overall setups differ substantially.  Compared with ChatTracker, our model improves AUC by **1.7% on LaSOT** and **7.2% on TNL2K**.  Compared with MambaVLT, our model improves AUC by **6.8% on LaSOT** and **0.3% on TNL2K**.
> Our work and **ATCTrack** were developed during the same period, and our manuscript was even made publicly available slightly earlier. Although there are indeed some superficial similarities between **ATSTrack** and **ATCTrack** (both segment sentences and have similar names), this is purely coincidental (and unfortunate for us). It would be unfair to directly compare our approach with **ATCTrack**, since it benefits from a stronger architecture (**AQATrack** vs. **ODTrack**), better initialization weights (**FastITPN** vs. original **HiViT**), larger training data (8 datasets vs. 4), and more instances per epoch (10k vs. 6k). As shown in **ATCTrack**’s original paper, the baseline of their method already demonstrates highly competitive performance.
>
> **2. Statement on novelty.**
> As mentioned previously, ATSTrack and ATCTrack are concurrent works, and our method was developed independently without being influenced by ATCTrack . The motivations of the two methods are fundamentally different. **ATCTrack** emphasizes the linguistic components most relevant to the target itself, training a model to identify these **key words** and align their semantics with visual-temporal cues. In contrast, our method seeks to fully exploit *all* components within the sentence. Specifically, we observe that different parts of a sentence convey information at distinct temporal and spatial scales, and based on this observation, we design an **attribute-specific method** to refine the features accordingly..
> Although VLToken is simple, it addresses an *important early-stage cross-modal interaction* issue that has been **overlooked by existing approaches**, and our ablation studies clearly show that this design is highly effective. Therefore, we believe its contribution remains meaningful.
>
> **3. Consistency of language decomposition.**
> In practice, decomposing language descriptions into different attributes is straightforward, and LLMs rarely make factual mistakes, making our method highly reproducible. We will release the decomposed text descriptions along with the code.
> The “errors” mentioned in the paper mainly refer to subjective inconsistencies. For example, based on our rule that “accessories are treated as appearance,” the phrase “head of a man” may be parsed as “category: head” and “appearance: of a man,” whereas a human annotator may intuitively regard the entire phrase as a “category.” Such cases are extremely rare and have negligible impact on tracking performance.
> We will clarify this phenomenon more explicitly in the revised manuscript.
>
> **4. Rationale for language decomposition.**
> As discussed in Sec. 3.2, the rationale behind decomposing language descriptions into attributes comes from a simple observation:  Descriptions about a target’s intrinsic properties correspond to only a small part of the image, while motion-related properties usually require temporal observation to determine (e.g., dancing vs. exercising).  Existing approaches attempt to adapt language using only the features from the global search region, which contains large amounts of background, lacks temporal cues, and does not explicitly localize the target—thus making such adaptations insufficient.  We believe this observation is intuitive and does not rely on complex theoretical assumptions. We have conducted extensive ablation experiments to quantitatively validate the effectiveness of our design.
> If you could clarify the type of justification you expect, or specify where you believe our motivation may be insufficient, we would be very happy to engage in further discussion to resolve your concerns.
>
> Finally, we will add the experiments on MGIT as soon as possible.

---

### Note · Authors · 2025-11-28

I have read and agree with the venue's withdrawal policy on behalf of myself and my co-authors.